

# Physiological effects and technical strategies of LED supplemental lighting for pitaya cultivation: a review

Ren Chen[1], Yiming Ding[1], Wenke Liu[1,2,3], Xianwei Zhan[1], Kexin Lin[1], Kaifeng Lian[1], Weilong Chen[1], Keyu Wang[1] and Shangfei Lin[1]

[1] Guangdong-Hong Kong-Macao Joint Laboratory for Intelligent Micro-Nano Optoelectronic Technology, School of Physics and Optoelectronic Engineering, Foshan University, FoShan, Guangdong, China
[2] Institute of Environment and Sustainable Development in Agriculture, Chinese Academy of Agricultural Sciences, Beijing, China
[3] Key Lab of Energy Conservation and Waste Management of Agricultural Structures, Ministry of Agriculture and Rural Affairs, Beijing, China

Corresponding author
Wenke Liu, liuwenke@caas.cn

## ABSTRACT

Pitaya is a high-value perennial tropical fruit known for its nutritional and health benefits. It is now widely cultivated in many tropical and subtropical countries, offering strong economic returns. China ranks first globally in pitaya cultivation, which includes both open-field production in tropical and subtropical regions and facility-based cultivation in temperate zones. As a long-day, light-loving plant, pitaya can be produced year-round. However, during off-season cultivation in winter and spring, weak light conditions and limited daylight hours lead to low flowering and fruit-setting rates, resulting in reduced yield and quality, factors that significantly constrain the industry's development. The core technological challenge in achieving high-quality, high-yield, multi-cropping pitaya production lies in inducing abundant, high-quality blooms in a staged manner using LED artificial lighting. Based on current domestic and international research on the physiological mechanisms and technical strategies of light-induced flowering in pitaya, the effects of LED light supplementation on flowering and yield, along with relevant technical parameters, have been clarified. Practical applications have demonstrated the feasibility of using three-dimensional, precise LED supplemental lighting to regulate flowering and photosynthesis in both facility and open-field cultivation. This technology synergistically promotes both vegetative and reproductive growth, significantly improving flowering and fruit-setting rates, increasing single fruit weight, enhancing yield and quality, and boosting annual production efficiency. This article comprehensively summarizes the enhancement effects and physiological mechanisms of LED supplemental lighting on pitaya flowering regulation, focusing on light intensity, light quality, and photoperiod, within the context of international research. It also analyzes existing challenges and proposes strategies such as optimizing LED light source design, accurately planning supplemental light periods and durations, and establishing three-dimensional lighting methods. These strategies aim to improve the efficiency of LED lighting systems and provide a theoretical foundation for developing a precise and efficient pitaya LED supplemental lighting technology system. In conclusion, LED supplemental lighting promotes both the quantity and timing of pitaya flowering, as well as fruit yield and quality. Red, blue, and far-red

light, combined with a photoperiod of 4–6 h, are recommended for effective application.

# INTRODUCTION

Pitaya is a perennial, trailing and pulpy shrub belonging to Cactaceae. Pitaya, a tropical plant with a triangular fleshy stem, has stem nodes that can climb and grow (*Li, 2014*). Pitaya is an emerging tropical and subtropical fruit tree that has attracted widespread attention in recent years. Due to its strong adaptability and unique flavor, its planting area has spread to tropical and subtropical regions all over the world. There are varieties such as red-skin and white-meat, red-skin and red-meat, and yellow-skin and white-meat (*Attar et al., 2022*). In Asia, it is mainly planted in China, Vietnam, Thailand, Malaysia, the Philippines and Indonesia; in the Americas, in Mexico, Colombia and Brazil; in Africa, in South Africa; and in Europe, in Spain. Since the 1990s, it has been successfully introduced in Taiwan, as well as Guangdong, Guangxi and Hainan provinces (*Jiang et al., 2011*). In recent years, the cultivation of pitaya has achieved remarkable economic benefits. The planting area in China, Brazil, Turkey and other countries has shown an increasing trend (*Rabelo et al., 2020*). Fruit farmers are willing to adopt new planting technologies and continuously increase production investment. By 2019, the planting area of pitaya in China had reached 66,700 hm$^2$, surpassing Vietnam and ranking first in the world (*Wang et al., 2024*). According to an article in China Fruit News (*Zhao, 2023*), which states that China ranks first in the world in dragon fruit production, the production of pitaya in China had exceeded 1.6 million tons by 2022. The main production areas are concentrated in tropical and subtropical provinces such as Guangxi, Guangdong, Guizhou, Yunnan, Hainan and Taiwan, of which the planting area of Guangdong and Guangxi accounts for 70% of the total area of the country (*Zhao, 2023*). In the vast southern region of China, Nanning City and Long'an County in Guangxi, and Zhanjiang City and Heyuan City in Guangdong Province are very representative in the field of pitaya planting. In recent years, facility pitaya cultivation in temperate regions has flourished in China, with a rapid development momentum. There is a coordinated development between facility cultivation in temperate regions and open-field cultivation in tropical regions, for example, in terms of the exchange of cultivation techniques and sharing of high-quality varieties, which jointly promotes the development of the pitaya industry.

Pitaya, a tropical fruit highly favored by consumers, is rich in nutrients and functional substances. It combines functions such as fresh-food consumption, processing, ornamentation, and medicine-making, thereby boasting high nutritional, economic, medicinal, and health-care values. *Lu, Zhu & Liao (2024)* analyzed the research and application progress of lycopene from pitaya. The study found that the beet red pigment in pitaya is of great significance. It has high hydrophilicity and coloring strength, and is

composed of eight monomers. Due to the aromatic ring and a large number of hydroxyl groups in the structure, it has good antioxidant and free-radical-scavenging ability. It can be used as an important natural colorant in the fields of food, medicine, and cosmetics. In the food industry, it can also act as a natural antioxidant, bacteriostatic agent, and indicator. In addition, pitaya is rich in plant protein, beet pigment, and water-soluble dietary fiber. These substances have special effects on human health. They can maintain physiological function, enhance immunity, and promote intestinal health (*Ferreira et al., 2023*). In addition, pitaya has a large amount of protein, fiber, carotene, Vitamin B, Ascorbic acid, glucose and a large number of trace elements (*Ferreira et al., 2023*). Moreover, pitaya has fewer calories (*Huang, Zhang & Hong, 2012*; *Sun et al., 2022*), which is a low-calorie and high-nutritional fruit. Pitaya flower also has health functions. The flower contains vitamin E, fatty acids, β-sitosterol, rapeseed sterol, flavonoids and triterpenoids. Its pharmacological effects involve bacteriostasis, anti-inflammation, enhancing immunity and reducing blood lipids. In addition, pitaya stems have positive effects on cough, colon cancer and antioxidation (*Huang, Zhang & Hong, 2012*; *Sun et al., 2022*).

Pitaya, a fruit with unique biological characteristics and significant industrial advantages, has gained increasing prominence in the global agricultural industry. A comprehensive exploration of pitaya's growth, development, cultivation techniques, and market potential is essential to promote the sustainable development of the pitaya industry, improve agricultural efficiency, and increase farmers' income. This is the primary motivation for writing this review. Pitaya boasts remarkable biological characteristics that provide significant advantages for its cultivation. These include light and shade tolerance, heat and drought resistance, as well as tolerance to poor soil and low-fertility fertilizers. Pitaya plants survive under these conditions, but under commercial conditions adequate irrigation and plant nutrition conditions are required. Pitaya has strong adaptability and can grow well in daily environments, so under optimized field management conditions and good water and fertilizer management, it can achieve high yields. Pitaya has strong environmental adaptability in tropical and subtropical regions, and can also adapt to adverse environments, so it can be widely commercialized for cultivation. These traits enable pitaya to thrive in diverse natural environments, demonstrating strong adaptability and laying a solid foundation for its widespread cultivation. Despite its strong adaptability, when it comes to achieving high-quality and high-yield production, light becomes a crucial factor for pitaya. Natural light conditions often fail to meet the plant's requirements, especially in terms of intensity and duration, which leads to the necessity of artificial lighting supplementation. From an industrial development perspective, pitaya offers additional advantages. The short flowering-to-harvest period significantly curtails the production cycle and improves efficiency. Furthermore, multiple harvests per year increase yield, while the high economic return provides substantial benefits to farmers. As living standards improve and the demand for healthy fruits grows, market demand for pitaya continues to rise. These factors make pitaya an ideal crop for industrial adjustment and poverty alleviation in China, as well as a promising candidate for developing protected fruit orchards. Many scholars have contributed valuable insights into pitaya's growth and

development. *Zhang et al. (2022)* discovered that during the critical period of flower and fruit development, the four endogenous hormones work together to finely regulate the entire development process of pitaya flowers and fruits. Moreover, the flowering period of pitaya can be precisely regulated by spraying exogenous hormones during its growth. This finding provides a theoretical foundation and technical support for regulating pitaya's production cycle and ensuring a consistent annual supply. Pitaya is a light-loving, long-day plant that has strict requirements for light intensity and duration in order to achieve high quality and yield. Light quality significantly influences pitaya's nutritional growth by affecting photosynthetic products. It also plays a key role in determining the timing of flowering, fruit-setting, and fruit quality. *Meng & Kramer (2024)* further demonstrated that extending night light duration can effectively accelerate the flowering process in long-day plants. This discovery provides valuable insights for pitaya cultivation and management, offering strong theoretical support for regulating flowering periods and promoting flowering through artificial light supplementation. However, pitaya's unique night-blooming characteristic has limited research on its reproductive phenology (*Kishore, 2016*). This gap in knowledge hinders a deeper understanding of pitaya's reproductive process, which in turn limits further development of the pitaya industry. Exploring the relationship between phenology, light quality, flowering quality, and economic benefits has become a key topic in current international pitaya research. In this context, the key question arises: How exactly does LED supplemental light affect the physiological processes of pitaya, and what are the optimal technology strategies for its application in pitaya cultivation? This review aims to delve deeply into these aspects to provide a comprehensive understanding and practical guidance for the industry. Investigating these relationships is expected to reveal the underlying mechanisms of pitaya's reproductive development, providing a scientific foundation for improving yield and quality. Moreover, due to the uncertainties and limitations of natural light conditions in pitaya cultivation, compensating for insufficient sunlight with artificial lighting has become a crucial technical requirement. This not only impacts yield and quality but also directly affects farmers' economic benefits and the sustainable development of the industry. In view of the importance of artificial lighting in pitaya cultivation, numerous studies have been conducted in this area. However, the existing research results are scattered, and a systematic review that integrates different viewpoints is needed to better guide the development of the industry. Additionally, *Shah et al. (2023)* systematically summarized international research progress on pitaya in nutrition, biology, and biotechnology, offering a valuable reference for future studies. They comprehensively reviewed various aspects of pitaya cultivation but didn't directly focus on LED lighting. However, they did mention that pitaya is a light-loving plant and supplemental light can regulate its flowering. This implies that LED lighting, a common form of artificial light, can play a role. Future research can build on this by exploring the optimal LED light spectra, intensity, and duration for pitaya growth and flowering based on the plant's physiological needs. *Oliveira & Tel-Zur (2022)* reviewed pitaya fruit production and orchard management, offering practical insights for improving yield and fruit quality, and promoting the widespread adoption of planting schemes. They didn't directly focus on LED lighting in pitaya

cultivation. However, they noted that pitaya was photoperiod-dependent, with optimal flowering and fruiting under a 12-h critical day length. They mentioned artificial light for off-season cropping, like in Taiwan where 6-h supplemental light (fluorescent lamps) can induce off-season production. This implies LED lighting can be explored for similar purposes, but its specific use and effects remain unstudied. In addition to the aforementioned studies, recent research by *Wang et al. (2024)* had also focused on the promotion of pitaya, flowering by supplementary light, revealing the mechanism by which light quality affects early pitaya ripening. Meanwhile, *Xie et al. (2022)* explored the role of LED supplemental light in the synthesis and metabolism of pitaya hormones, and found that an appropriate red to blue light ratio can enhance the hormone synthesis of pitaya, thereby affecting fruit quality and yield. These and other studies further deepen our understanding of the complex relationships in pitaya cultivation under LED assisted light. However, there is currently a lack of comprehensive research that combines these various aspects to provide growers and researchers with a unified perspective. In summary, the pitaya industry faces significant opportunities and challenges in its development. This review aims to thoroughly summarize research in related fields, analyze current research hotspots and challenges, and offer clear directions for future studies, while providing scientific and practical guidance for the sustainable development of the pitaya industry.

Our objective is to comprehensively analyze the existing research results, thus clarifying the current situation of LED supplementary light's physiological effects and technology strategy in pitaya cultivation. The audience for this literature review encompasses horticulturists, pitaya growers, agricultural researchers, students and academics in agricultural and plant sciences, as well as industry stakeholders. Horticulturists and growers seek to boost production efficiency and fruit quality; agricultural researchers can identify research gaps and promote cross-disciplinary studies; students and academics peers can use it for learning and teaching and learn the most advanced and summarized information on the current research on pitaya nourishing light through this article; graduate students can read articles as a quick reference to understand the industry; Industry stakeholders, like LED lighting manufacturers and those in the pitaya value-chain, can develop better products and marketing strategies; fruit farmers and agricultural technology extension personnel can directly obtain technical information on supplemental light strategies from the articles. Meanwhile, this article can promote the development of the pitaya industry in the world, especially in China. This review aims to offer a thorough understanding of LED supplementary light in pitaya cultivation, including its advantages and areas for improvement in related technologies.

## SURVEY METHODOLOGY

In this review, we employed a meticulous approach to ensure comprehensive and unbiased coverage of the literature, aiming to provide a well-rounded understanding of the physiological effects and technology strategy of LED supplemental light for pitaya cultivation. The main search tools used in this study are Google Scholar, China National Knowledge Infrastructure (CNKI) database, Web of Science, and Baidu Scholar. Google Scholar offers a vast collection of academic resources, enabling us to access a wide range of

international studies, including journal articles, conference articles, and reports. CNKI, on the other hand, is a valuable source for Chinese-language literature, which is crucial for capturing research specific to China's pitaya cultivation. We carefully formulated search terms to ensure precision and comprehensiveness. The search terms used in this study were "pitaya", "LED supplementary light", "flowering", "fruit yield", "nutritional quality", "photoperiod", "light intensity", "light quality", and "three-dimensional supplementary light". These terms were combined in various ways, such as "pitaya and LED supplementary light AND flowering" and "LED supplementary light AND fruit yield AND pitaya", to retrieve relevant articles effectively.

Articles were selected based on specific inclusion criteria. Firstly, they had to be directly related to the use of LED supplementary light in pitaya cultivation. This meant that studies focusing on general plant lighting without specific reference to pitaya or those not involving LED light were excluded. Secondly, we prioritized articles published within the last 5–10 years to incorporate the latest research findings. However, some classic and highly-cited older publications were also included as they provided essential theoretical frameworks or references. Articles were excluded if they did not meet the relevance criteria. For example, articles that were not accessible in full-text, despite attempts to obtain them through institutional subscriptions or other means, were removed from the review. Additionally, studies that did not contribute to the understanding of the physiological effects or technical strategies of LED supplementary light for pitaya were not considered.

To achieve comprehensiveness, we employed multiple search strategies. In addition to the basic keyword searches, we utilized the "Related Articles" and "Cited By" features on Google Scholar. This allowed us to discover additional relevant studies that might not have been retrieved through the initial keyword search. From CNKI, we used the database's advanced search options, such as searching within specific journals known for publishing high-quality pitaya-related research. To maintain an unbiased review, we made sure to select articles from a diverse range of sources. We included articles with different research focuses, such as those exploring the impact of LED light on pitaya flowering, fruit yield, and nutritional quality. Articles with both positive and negative findings regarding LED supplementary light were also incorporated to present a balanced view. Moreover, we covered research from different geographical regions, including China, Southeast Asia, and other areas where pitaya is cultivated, to account for regional differences in cultivation practices and environmental conditions.

During the review process, we identified several gaps in the existing literature. There is a lack of in-depth research on the long-term effects of LED supplementary light on pitaya plants. For instance, the impact of continuous use of LED lighting on the genetic and physiological characteristics of pitaya over multiple generations remains unclear. Additionally, the optimal LED lighting conditions for different pitaya varieties and growth stages are not fully understood. Future research could focus on filling these gaps. Long-term field experiments could be conducted to monitor the long-term impacts of LED lighting on pitaya plants. Research on the interaction between LED lighting and other environmental factors, such as temperature, humidity, and soil nutrients, is also needed.

Moreover, there is a great potential for cross-disciplinary collaborations. For example, collaborations between plant physiologists, who can study the biological mechanisms of pitaya's response to LED light, and electrical engineers, who can develop more efficient and customized LED lighting systems, could lead to significant advancements in pitaya cultivation technology.

## LIGHT REQUIREMENT AND PHOTOPHENOLOGICAL CHARACTERISTICS OF PITAYA FLOWERING

Pitaya, a tropical fruit with soaring popularity, owes much of its growth and development to light. It is a light-loving, long-day plant. Its flowering is intricately linked to specific light conditions. Sufficient light intensity and duration are crucial for its vegetative and reproductive growth, especially for timely flowering induction. Understanding the light requirement and photophenological characteristics of pitaya flowering are crucial. It not only unlocks the secrets to optimizing its yield and quality but also guides growers in implementing precise lighting strategies. Let's explore these aspects in detail.

The daily light integral (DLI) refers to the total amount of photosynthetically active radiation (PAR, 400–700 nm) that plants receive per square meter per day, measured in $mol \cdot m^{-2} \cdot d^{-1}$. It's a key metric in plant lighting, influencing growth, flowering, and fruiting, including in crops like pitaya. DLI combines two key factors-light intensity and duration-that are crucial for crop photosynthesis, yield, and quality. Plants with a DLI of 3-6 $mol \cdot m^{-2} \cdot d^{-1}$ are shade-tolerant, 6–12 $mol \cdot m^{-2} \cdot d^{-1}$ are moderate-light, 12–18 $mol \cdot m^{-2} \cdot d^{-1}$ are photophilic, and those with more than 18 $mol \cdot m^{-2} \cdot d^{-1}$ are strong-light plants. Pitaya, as a light-loving plant, has specific DLI requirements. For example, the optimal DLI for cut roses is 22–30 $mol \cdot m^{-2} \cdot d^{-1}$. Pitaya requires supplemental light when its DLI is below 12 $mol \cdot m^{-2} \cdot d^{-1}$ to achieve high yield and quality. Both light intensity and photoperiod are crucial for attaining an appropriate DLI. The critical day length for pitaya flowering is approximately 12 h (*Jiang et al., 2012*), and photoperiods ensure the transition from vegetative to reproductive growth. However, from November to March in the northern hemisphere, sunlight is weaker and day length shorter. Due to the lack of sunlight, open-pollinated pitaya may fail to flower or produce poor-quality fruit. Although temperatures in controlled environments are higher than in open fields, light intensity is lower and duration shorter, resulting in more challenging light conditions during flowering. Therefore, increasing light intensity and extending photoperiods can promote flower bud differentiation and induce timely, high-quality flowering.

In addition to light requirements, temperature also plays a vital role in pitaya growth. The optimal temperature range for pitaya growth is 25 °C to 35 °C. If temperatures fall below 10 °C or exceed 38 °C, pitaya growth halts. Except for Hainan, winter temperatures in China are low, and sunshine duration is short from November to March. In practice, the lighting requirements for pitaya are well-defined. Pitaya typically flowers and fruits in multiple batches per year under suitable growth conditions, with flower bud differentiation controlled by light and temperature. When cultivated in open fields, the flowering period usually lasts from May to November, the fruit maturity period from June to December, and the production gap spans from January to June. In South China, flowering is typically

induced in February, with harvests from April to May, and again in October, with harvests in December. To prolong the fruit production period of pitaya and reduce the production gap, it is necessary to adjust temperature and light conditions during off-season cultivation in winter and spring to induce flowering and fruiting. Greenhouse cultivation allows pitaya to grow in temperate regions during the winter and spring low-temperature period, and flowering regulation techniques can enable off-season production (*Hu et al., 2017*). Studies show that light affects flower bud sprouting in pitaya (*Yi, Liu & Wei, 2004*; *Yan & Lin, 2000*). In southern China, insufficient sunlight during winter and spring limits both vegetative and reproductive growth, particularly flowering. Poor flowering and fruit setting lead to reduced yield and quality, lowering the production efficiency for farmers. Flowering marks the transition from vegetative to reproductive growth in fruit trees, and it has been confirmed that the photoperiod for pitaya flowering must exceed 12 h for blooming to occur (*Xiong et al., 2020*). Studies have shown that the use of light compensation and flower control technologies can advance or extend the flowering period of pitaya, thereby improving yield and staggering harvests (*Saradhuldhat, Kaewsongsang & Suvittawa, 2009*; *Chen et al., 2019a*). Therefore, artificial light supplementation to enhance light intensity and duration is the key technology for ensuring abundant, high-quality flowering, multiple cropping, and high yield. This approach is applicable not only in southern China, but also in northern China's facility-based pitaya cultivation, where light deficiency in winter and spring is more severe, making light supplementation even more crucial. Internationally, leading research institutions in the field of pitaya light environment regulation and supplementation include Hainan University, National Chiayi University, National Chung Hsin University, and Taiwan University. Research on long-day plant light environment and flowering induction is conducted at institutions such as the University of Delaware and Michigan State University. *Li et al. (2024b)* reviewed various nighttime lighting methods for pitaya, including light quality, duration, lamp height, and hanging methods.

To date, there has been no comprehensive review on the physiology of pitaya flowering induced by supplemental light and LED light supplementation strategies. This article provides a thorough summary of the effects and physiological mechanisms of LED supplemental light on pitaya, focusing on light intensity, light quality, and photoperiod control in flowering. Based on this, strategies are proposed to enhance LED light supplementation efficiency, including optimizing LED light source design, accurately planning light supplementation periods and durations, and establishing three-dimensional light supplementation methods. These strategies aim to provide a theoretical foundation for developing an accurate and efficient LED light supplementation system for pitaya.

## PHYSIOLOGICAL MECHANISM OF LED INDUCED FLOWERING IN PITAYA

LED supplemental light plays a crucial role in pitaya cultivation, especially in inducing flowering. Exploring the "physiological mechanism of LED induced flowering in pitaya" is of great significance. It not only allows us to deeply understand the principles of pitaya's response to light but also provides a theoretical basis for growers to precisely regulate the flowering period. Now, let's delve into the physiological mysteries behind-induced

flowering in pitaya. is the optimal light source for pitaya lighting. Compared to traditional light sources, it offers significant advantages in lamp shape, weight, power ratings, spectral composition, luminous surface, and optical design and management. It provides higher light efficiency, better energy savings, and meets the demands of modern agricultural development. As a semiconductor solid-state light source, LED has been widely used in plant production, particularly in artificial plant factories and greenhouse horticulture, with excellent results. The main advantages of using LED for pitaya supplemental light are: (1) It is a cold light source, energy-efficient, long-lasting, and effective for close-range light supplementation to the pitaya canopy. (2) The LED spectrum is adjustable, allowing for tailored light conditions to match pitaya's growth stages and maximize the physiological effects of light quality. (3) LED light sources are compact, flexible for installation, and suitable for three-dimensional lighting. *Liu (2021a)* studied stereoscopic lighting and its application strategies in plant factories. It can irradiate from multiple angles, including the top of the greenhouse, the upper canopy, the sides of the canopy, the interior of the canopy, and the lower canopy, providing omni-directional lighting for pitaya growth. This enhances the canopy's light interception and utilization, maximizing light efficiency to achieve high quality and yield. (4) The LED light source is adjustable and controllable, enabling dynamic lighting, such as intermittent, pulse, continuous, and alternating light, which sunlight cannot provide. By adjusting light quality and intensity fluctuations, it can stimulate the plant's photophysiological potential, maximizing photosynthetic efficiency per unit of light energy.

## Light intensity

Light intensity is a crucial factor in pitaya growth. It significantly impacts photosynthesis, flowering, and fruiting. Let's explore how different light intensities affect pitaya and why it's essential for optimal cultivation. Light intensity plays a crucial role in inducing flowering, with strong light being particularly effective. Additionally, the effects of far-red light on flowering induction vary under different light conditions, with distinct physiological mechanisms. Under high DLI (greater than 19 mol·m$^{-2}$·d$^{-1}$), plants bloom faster, and the addition of far-red light (FR) has little effect. For example, the flowering time of ornamental plants under red + white (R + W) and red + white + far-red (R + W + FR) light is similar (*Garrett, Meng & Lopez, 2018*). In contrast, long-day plants (Petunia hybrida and goldfish grass) grown under low DLI bloomed earlier with the supplementation of 2 μmol·m$^{-2}$·s$^{-1}$ of far-red light (R + W + FR) compared to R+W alone. Therefore, under low DLI conditions (less than 6 mol·m$^{-2}$·d$^{-1}$), far-red light supplementation can promote the flowering of long-day plants. As is well known, when the precursor form (Pr) is stable, its concentration remains constant, while the photoactivated form (Pfr) is unstable and requires far-red light for its conversion. For short-day plants, the interruption of red light and dark periods inhibits flowering, whereas far-red light promotes it; conversely, for long-day plants, red light promotes flowering, and far-red light inhibits it. This reversible response to red and far-red light in flowering indicates the involvement of the phytochrome system in flower induction. Therefore, at the end of the light period, the Pfr/Pr ratio is high, and supplementation with strong light or high DLI

during the dark period can continue to convert Pr to Pfr, increasing the Pfr concentration and enhancing the Pfr/Pr ratio. Under the condition of weak light supplementation during the dark period, the conversion of Pr to Pfr was minimal, and Pfr maintained its self-degradation rate under light conditions, leading to a slow decline in the Pfr/Pr ratio. The mechanism of far-red light supplementation differs under low and high DLI (*Kohyama, Whitman & Runkle, 2014*). At low light intensity ($<10$ μmol·m$^{-2}$·d$^{-1}$), far-red radiation during the photoperiod is more important for flowering induction, whereas at high light intensity, this effect is less significant (*Kohyama, Whitman & Runkle, 2014*). There are two modes of light-dependent flowering control: (1) photosynthesis under high-intensity red-rich radiation and (2) flowering controlled by phytochrome under low-intensity far-red radiation. At high light intensity, increased photosynthesis is the primary mode for inducing flowering, while low-intensity far-red light-mediated flowering promotion *via* phytochrome is secondary. In contrast, under low DLI, the effect of photosynthesis on flowering is significantly reduced, and the phytochrome-mediated acceleration of flowering *via* low-intensity far-red light becomes the dominant mode. Under low DLI, supplementary far-red radiation can reduce the R/FR ratio and induce the shade avoidance response, thus accelerating flowering (*Franklin & Whitelam, 2005*). Therefore, 400–700 nm photosynthetically active radiation with a DLI intensity of 5–20 mol·m$^{-2}$·d$^{-1}$ can accelerate flowering and increase the number of flowering plants and dry matter (*Faust et al., 2005*; *Moccaldi & Runkle, 2007*).

Phytochrome regulates flowering and stem elongation by receiving light signals (minimum 1–2 μmol·m$^{-2}$·s$^{-1}$) (*Thomas & Vince-Prue, 1997*; *Whitman et al., 1998*). Short-day plants require a low Pfr/Pr ratio, while long-day plants require a high Pfr/Pr ratio. At the end of the photoperiod, the phytochrome mainly exists in the Pfr form due to the dominance of red light, resulting in a high Pfr/Pr ratio. After entering the dark period, the Pfr/Pr ratio decreases as Pfr gradually converts to Pr or degrades. Once the ratio reaches a certain threshold with the extension of the dark period, it promotes the formation of flower-inducing substances and triggers flowering. Chrysanthemum, a short-day plant, uses shading to shorten the light period and promote flowering. Long-day plants, on the other hand, require a short dark period to prevent the reduction of Pfr content and can flower under continuous light. If the dark period is interrupted by red light, the increase in Pfr/Pr ratio can inhibit the flowering of short-day plants and promote the flowering of long-day plants. Long-sunshine plants, such as pitaya, rhododendron, and camellia, can flower earlier or later by adjusting the light duration. Thus, long-day plants flower most quickly under suitable red and far-red light irradiation. In conclusion, Pfr and Pr are receptors for far-red and red light, respectively, and their content is negatively correlated with the dose of far-red and red light, being reversible and mutable. Flowering is determined not by the absolute amount of Pr and Pfr, but by the ratio of Pfr/Pr. Sufficient light duration, high light intensity, and low light intensity with far-red light supplementation are essential for inducing flowering in long-day plants.

Pitaya can improve both the quantity and quality of flowering through supplemental light, and the impact on yield increase has been well-documented. *Pang & Chen (2024)* demonstrated in their study on off-season artificial light supplementation technology for

dragon fruit flowering that nighttime supplemental light significantly enhances the flowering rate and fruit quality of pitaya. *Lai et al. (2018)* used 2-year-old short-core red-flesh pitaya plants as test subjects and found that adding light for 6 h every night over 45 consecutive days effectively promoted flowering in winter within greenhouses. *Wang et al. (2024)* applied 15 W LED red and blue light at a 7:1 ratio from 6:30 to 11:30 p.m. and observed that nighttime supplemental light in winter interacted with endogenous hormone signals, especially cytokinins, to regulate flower bud formation (*Wang et al., 2024*). Under high temperature conditions, pitaya adapts to strong light irradiation, and while shading can regulate its physiology, it has a limited effect on yield. *Almeida et al. (2021)* investigated the effects of five different light intensities (full day, 35%, 50%, 65%, and 80% shading) on the morphological, biochemical, physiological, and anatomical characteristics, as well as the yield of pitaya. The results showed that light conditions significantly impacted the growth, anatomical structure, photosynthetic pigments, gas exchange, yield, and the correlation of red pitaya (*Almeida et al., 2021*). Pitaya exhibits high phenotypic plasticity, enabling it to fully adapt to cultivation environments under full sunlight, thereby achieving the highest yield. The study also suggested that pitaya should be cultivated under full sunlight or with 35% shading, as these conditions optimized its eco-physiological variables and yield. *Li et al. (2024a)* identified 59 differentially expressed hub-like basic helix-loop-helix (hubHLH) genes during winter light supplementation-induced flowering. Gene ontology (GO) analysis revealed that these genes were enriched in functions related to the response to red light or far red light, light stimulation, sexual reproduction, and radiation response. These findings suggest that the hubHLH gene family may play a role in the winter light supplement-induced flowering of pitaya (*Li et al., 2024a*). *Chien et al. (2024)* examined the effects of off-season cultivation net rooms and light sources on the quality and yield of pitaya during spring and winter. In the spring and winter of 2017–2018, *Chien et al. (2024)* conducted a study in which mature 6-year-old pitaya plants were grown in both net room and outdoor environments. The study utilized 100 W incandescent bulbs (IBS) and 23 W compact fluorescent bulbs (CFBs) for artificial lighting to achieve night interruption and assessed the flowering rate and fruit production. The results showed that in spring, the combination of net room + IBS and net room + CFBs induced flower bud emergence at least 2 weeks earlier than other treatments in the control group. The flowering frequency was highest with net room + IBS, with four flowering cycles in spring and three in winter. In all treatments, nighttime artificial lighting in spring significantly increased yield, with IBS and CFBs showing similar effects. Regarding fruit quality, outdoor planting in spring yielded better results than indoor planting, while in winter, net room planting, especially with IBS, was more effective in enhancing yield. Overall, net house planting demonstrated significant advantages for off-season production, including earlier peak flowering in spring, more flowering cycles in both spring and winter, and higher flowering rates in winter.

## Light quality

Many studies have shown that light quality can regulate plant morphology, yield and quality, and light quality can also regulate flowering induced by light supplement. Light

quality is a vital element in pitaya cultivation. It varies with different spectra and affects the plant's physiological processes. Different light quality have varied effects. We found that red and blue light combinations promote flower bud differentiation, increase yield, and improve fruit quality. Besides, white light is good for soluble protein content, and blue light for soluble sugar. Far red light can efficiently induce flowering, and green light may play a role in high-intensity configurations. *Liu & Cha (2019)* comprehensively and deeply elaborated on plant light quality physiology in their book plant factory plant light quality physiology and its regulation (*Liu & Cha, 2019*). So far, the research on the quality of LED supplemental light has mostly focused on red, blue, white and green light and their proportion screening optimization. *Chen et al. (2019b)* took red meat pitaya as the test material, and set the control without supplemental light, LED white light, blue-red light (the proportions were 1:8, 1:6, 1:4, 1:3, 1: 2, respectively). For providing plant supplemental lighting, LED lamps were hung at the heights of 50 cm and 70 cm, and the effects of light supplemental quality and lamp hanging height on flower bud differentiation, phenology period and fruit quality of pitaya were studied to screen out LED light supplemental quality that can promote early bud emergence in spring, increase the number of flower buds and improve fruit quality (*Chen et al., 2019c*). The results showed that the number of flower buds differentiated under supplementary light treatment was more than that without supplementary light treatment. When the ratio of blue light to red light was 1:4 and the hanging height was 50 cm, the number of flower buds differentiated and the yield were the highest, with an average yield of 2.236 kg/plant. Under the treatment of blue and red light ratio 1:2 and light source height 50 cm, the soluble solid content and quality of fruit were relatively the highest. *You et al. (2021)* studied the effects of light quality supplemental light of nine LED red and blue light ratios and red green and blue light ratios (lamp spacing of 1.5 m, lamp to plant distance of 50 cm) on the flowering number and yield of pitaya plants. The results showed that the red, green and blue light treatment (23:75:2) could not only obtain more flower buds and higher flower branch formation rate, but also make pitaya bloom earlier and increase the yield per plant. *Xie et al. (2022)* studied the effects of red light, white light and blue light on the physiological characteristics of pitaya during night replenishment. The results showed that all three kinds of light supplementation could increase the soluble protein content of pitaya fruit plants, and white light had the best effect. The content of soluble sugar in Pitaya plants could be significantly increased by the three kinds of supplementary light, and the blue light was the best. Supplementation of white light and blue light can significantly reduce cytokinins (CTK) content, supplementation of white light and red light can significantly increase acetic acid (IAA) content, and supplementation of red light and blue light can significantly increase gibberellic acid (GA) content. Red and white light supplementation can significantly increase flowering rate, and red light supplementation can significantly increase fruit yield. In addition, *Park & Runkle (2019)* found that moderate intensity blue light (80 $\mu mol \cdot m^{-2} \cdot s^{-1}$) could weaken the effect of R:FR on extended growth, but had no significant effect on the role of FR in promoting flowering in long-day plants. The results showed that in the process of regulating plant expansion and growth, the blue light signaling mechanism mediated by cryptoanthocyanin played a leading role in the FR signaling

mechanism mediated by photoallergens, rather than flowering (*Park & Runkle, 2019*). *Chen et al. (2019c)* found that the regulation mechanism of supplemental light treatment with a blue to red light ratio of 1:2 on sugar accumulation in pitaya pulp may be that it inhibits the acid conversion (AI) activity in pitaya fruit, reduces the decomposition of sucrose to glucose and fructose, and regulates the activities of sucrose phosphate synthase and sucrose synthase synthesis, thereby promoting the accumulation of sucrose in fruit (*Nomura, Ide & Yonemoto, 2005*). *Liu (2021b)* found that both the supplemental time and spectrum of LED bulbs can significantly affect the flowering of pitaya, and the time required for budding of yellow light 15 W and combined yellow light 15 W (yellow light: red light = 10:1) is shorter than that of purple light 15 W. When the temperature exceeds 18 °C in spring, yellow light 15 W and combined yellow light 15 W can be used as the light source in the greenhouse, and the supplemental light time is 5 h/night.

The addition of far red light in the LED supplementary light quality is not only efficient in promoting the flowering induction of pitaya, but also has the advantage of ultra-low energy consumption. The conventional combination of red light, blue light, white light and green light can be divided into high intensity configuration and low intensity configuration. The high intensity configuration can only use red light, blue light, white light and green light and their combination to supplement light, while the low light intensity configuration must add far red light to the supplement light quality to induce flowering of pitaya fruit by shade avoidance effect. Green light may play an important role in the high-intensity configuration of supplementary light, because the green three-pointed branches of pitaya can absorb and utilize green light more efficiently than ordinary green leaves (*Terashima et al., 2009*). Blue light regulation of flowering requires a very high light intensity, above 30 $\mu mol \cdot m^{-2} \cdot s^{-1}$ (*Meng & Runkle, 2015*).

## Photoperiod

Photoperiod is one of the important factors affecting plant flowering, which has important effects on flower bud differentiation, fruit quality improvement and area yield. Photoperiod is an environmental light signal that regulates the process of flower bud dormancy and germination, root formation and flowering. Under short day conditions, photoperiodic light can be used to promote the flowering of long-day plants by exceeding the critical photoperiodic duration, and the shortening of photoperiodic shade can inhibit the flowering of long-day plants (such as rhododendrons and camellia). Similarly, shorter-day plants are encouraged to bloom by shading, while shorter-day plants, such as chrysanthemums, are inhibited by lengthening and supplementing. The photoperiod is a key determinant in pitaya's life cycle. It dictates flowering, fruiting, and overall plant health. Dive into how specific photoperiods influence pitaya and their implications for cultivation. Plant flowering is a complex physiological process, which is affected by a variety of signals, including exogenous factors such as photoperiod and temperature, and endogenous signals such as plant hormones (*Bao et al., 2020*; *Blázquez, Ahn & Weigel, 2003*; *Campos-Rivero et al., 2017*; *Wang, 2014*). Studies have shown that plant flowering is synergistically regulated by light and plant hormones (*Biswal & Panigrahi, 2020*). The process of flower formation in plants is divided into three stages: flower induction, flower

excitation and flower development. *Demers et al. (1998)* found that prolongation of photoperiod was beneficial to tomato growth (*Biswal & Panigrahi, 2020*). During the growing season from April to October, pitaya blooms for about 30 to 35 days, while in winter it usually blooms for 40 to 45 days. Therefore, winter and spring pitaya need to shorten the flowering period by supplemental the light at night.

*Jiang et al. (2012)* conducted a photoperiod regulation test for flower bud formation from June to December 2009, and a feasibility test for off-season production in 2011 (*Jiang et al., 2012*). It was found that shortening the sunshine duration to 8 h in summer could inhibit the development of flower buds at the distal end of stem and promote the germination at the proximal end of stem (*Nerd et al., 2022*). During the period of autumn equinox to winter solstice, night light treatment can effectively induce flower bud formation, and its critical day length is about 12 h. It can be concluded that the treatment of nighttime light interruption during the spring equinox from September to March of the following year can be used for the anti-season production of red meat pitana. Further study showed that the duration of night light interruption required for flower bud differentiation was longer in cooler season than in warmer season.

Dark period interruption is an effective method to prolong photoperiod and promote flowering by low light intensity supplementary light. Implementing dark phase interruption with red light and FR can stimulate phytochrome response, and red light and FR lead to morphologic conversion between Pfr and Pr, down-signaling, delaying or stimulating flowering (*Borthwick, Hendricks & Parker, 1952*; *Hendricks & Borthwick, 1967*). In short-day plants, FR scintillation increased the length of dark period and stimulated flowering. On the contrary, for long day plants, flashing red light in the dark period will prolong the photoperiod and stimulate flowering. This regulation is reversible, and short-day plants will not flower after dark FR and red scintillation, because plants have experienced a short dark period, and most of the photopigments are Pfr forms. Using low light intensity to interrupt light during dark period can stimulate early flowering and increase fruit yield (*Cao et al., 2016*). *Cao et al. (2016)* studied 20 $\mu mol \cdot m^{-2} \cdot s^{-1}$ red light dark phase interruption (20:00 to 8:00, each duration of 10 min, the frequency of light supplement is every 1, 2, 3 and 4 h), which increased the early yield of tomato and made the plants more compact and healthy (*Cao et al., 2016*). At present, the lack of FR spectrum in most greenhouse LED lighting leads to inverse shade avoidance response, that is, low plant compactness biomass, reduced leaf area, weak light interception ability, and lack of photosynthetic effect (*Kalaitzoglou et al., 2019*; *Zhen & Bugbee, 2020*). We have concluded the effects of LED supplemental light on pitaya flowering, yield and quality in Table 1.

## TECHNICAL STRATEGIES FOR INDUCING FLOWERING IN PITAYA FRUIT USING LED SUPPLEMENTAL LIGHT

### Types of supplemental light fixtures

LED lighting fixtures, with advantages such as adjustable power, tunable spectra, and controllable operation, are ideal for supplemental light applications. Typically, a combination of white, red, and far-red light is used to meet the specific growth needs of

**Table 1  Effects of LED supplemental light on pitaya flowering, yield and quality.**

| Lighting treatments | Main results | References |
|---|---|---|
| 100 W incandescent bulbs or 23 W compact fluorescent bulbs; photoperiod 4 h | Extended off-season flowering cycles; obtained larger size and higher sugar winter fruit | Chien et al. (2024) |
| Night-breaking treatment photoperiod 4 h | Promoted flowering in 77% tested clones cultivars | Wang et al. (2024) |
| Three supplemental light durations (18:30–22:30; 22:30–2:30; 2:30–6:30) | Improved number of flower buds, flowering branch rate and single batch yield best in the middle night; flowering in advance 15 days | Xu et al. (2023) |
| Spring 4 h; autumn 5 h | Promoted the differentiation of pitaya flower buds; increased the number of flower buds and yield; had no effect on soluble substances | Gan (2022) |
| Red: blue 7:1; photoperiod 5 h | Promoted flower bud formation and development; affected hormone synthesis and metabolism; Improved fruit quality and yield | Xie et al. (2022) |
| Five shading treatments (full light, 35%, 50%, 65%, 80% shading) | Increased chlorophyll content; maximized fruit yield under full light, ranked second under 35% shading; minimized yield under 80% shading | Almeida et al. (2021) |
| LEDs (35, 30, 24, 20 W) and 100 W incandescent bulbs; photoperiod (6, 8,10 h) | Promoted the flowering and fruiting of pitaya; improved yield and fruit quality | Rashid, Azam & Chowdhury (2021) |
| Nine red, green and blue spectra | Promoted the early flowering, yield per plant | You et al. (2021) |
| Four nightbreak durations in late autumn and early spring (4, 2, 1, 0.5 h) | In late autumn, the longer the night time, the more buds will bloom, and within 1–4 h, the number of buds can reach 50%; in late spring, less distinct of the duration effect, decreased the days to flowering shoots 50% as the nightbreak duration increased, 28 d in the 4 h | Jiang (2020) |
| Red: blue 7:1; light intensity 110 lx; photoperiod 5 h | Promoted flowering; affected gene expression; Regulated flowering—related transcription factors | Xiong et al. (2020) |
| Night-interruption lighting 4 h; Long-day conditions (14 h light, 10 h dark) | Induced off-season flowering under 14 h; Higher success in warmer temperatures (≥29/19 °C), no flowering below 27/17 °C; accelerated bud development and emergence at 32/22 °C; unaffected vegetative bud emergence by temperature | Chu & Chang (2020) |
| Blue: red (1:8, 1:6, 1:4, 1:3, 1:2), white light LED; photoperiod (2 h, 3 h after 10 days) | Promoted flower bud differentiation; Highest number of flower buds at 1:4 with height of 50 cm; maximized soluble solid content with the best sugar-acid ratio and optimal flavor at 1:2; highest glucose and fructose contents in 1:6; Highest citric acid content in 1:4 | Chen et al. (2019b) |
| Blue: red (1:4, 1:2); 18 W; photoperiod 2.5 h | Enhanced soluble solid content and sucrose content in mature fruits under the 1:2, with the optimal flavor; increased average single-plant yield | Chen et al. (2019c) |
| Light quality (red-yellow, yellow, red-blue light); 15 W; photoperiod 5 h in spring and autumn | Promoted flower bud differentiation; best effect with red-yellow light, followed by yellow light, and red-blue light; Spring flower induction significantly better than autumn. | Chen et al. (2019a) |
| Light quality (far red, red, blue); photoperiod (16 h, 9 h) | Shortened flowering time, with an earlier fruit ripening period in 9 h or far red regulation; accelerated the flowering process with far red light at night at low DLI; Increased weight in single fruit at high DLI; got higher single fruit weight with red: far red 3:1; improves fruit uniformity with blue light | Owen, Meng & Lopez (2018) |
| Red + white or red + white + far red LED; photoperiods (10, 13, 16 h) | Long day plants bloomed faster at low DLI; promoted flowering at high DLI; Increased stem elongation in most species, prolonged exposure to far red radiation | Owen, Meng & Lopez (2018) |
| Photoperiod 6 h | Increased the rate of plants flower bud formation | Lai et al. (2018) |
| Red, blue, white light; light intensity 150–220 lx; photoperiod 4 h | Promoted effects on the contents of metabolites, chlorophyll, endogenous hormones in pitaya stems, the number of flowers, and fruit yield with red and white lights | Tran, Yen & Chen (2015) |
| Photoperiod (4 h in winter, 6 h in Off-season production) | Promoted flower bud formation in 4 h cut-off from the autumnal equinox to the winter solstice in September; achieved off-season production with 4 h cut-off treatment from September to the vernal equinox of the following March; promoted flowering with 4 weeks of light cut-off treatment in autumn | Jiang et al. (2012) |

plants (*Owen, Meng & Lopez, 2018*). When it comes to inducing flowering in pitaya with LED supplemental light, the choice of lighting fixtures matters. Different types offer unique features. Let's explore these fixture types and their impacts on pitaya flowering. Pitaya fruit LED supplemental light technology includes aspects such as LED fixtures, hanging methods, technical parameters, and control strategies. *Feng et al. (2013)* found that LED supplemental lights outperform the most commonly used sodium lights, offering advantages such as energy efficiency and ease of installation. The innovative design of LED fixtures covers multiple aspects, including the selection of fixture types (*e.g.*, bulb lights, UFO lights, strip lights, tube lights, floodlights, high-power square lights, and spotlights), power design, light-emitting surface and secondary optical design, spectral design, and control devices. The hanging methods include three options: top-down illumination, side illumination, and bottom-up illumination. The design goals for the fixtures are as follows: (1) The shape and size should be appropriate without blocking sunlight; (2) the light coverage should be strip-shaped, with a suspension height of 50 cm to cover each row of the canopy and reduce inter-row light loss; (3) the spectrum and light intensity should be appropriate, providing even illumination with energy efficiency; (4) the photoperiod should be controllable, focusing on weak light with the use of dark-period interruption techniques. It is recommended to use high-light-efficiency, high-luminous-flux LED lights for supplemental light, which are more energy-efficient, safer, and more controllable than high-pressure sodium lights. During the off-season production of pitaya, effective light-induced flower stimulation plays a crucial role in saving energy and maintaining stable productivity. *Nguyen et al. (2021)* investigated the use of lighting by Vietnamese pitaya farmers in the artificial flowering induction process (*Liu et al., 2019*). The study found that incandescent lights of 75-100 W are commonly used in pitaya cultivation for artificial flowering induction in places such as Taiwan, Thailand, and Vietnam, where 4 to 10 h of lighting per night is required, leading to high energy consumption. In Vietnam, it was found that only about 10% of farmers used 20 W CFL lights, but these were not widely adopted due to issues such as long germination times and low flowering efficiency. Given that the emission spectrum of the currently used energy-saving lamps does not align with the absorption spectrum of photoreceptors involved in pitaya flowering, *Nguyen et al. (2021)* proposed three improved energy-saving lamps with a power capacity of 20 W, whose emission spectra focus on red and far-red light (*Liu et al., 2019*). Compared to incandescent lamps (60 W), the improved energy-saving lamps showed relatively better performance in increasing the number of flower stems, bulb count, fruit number per plant, and fruit yield. The recent success of commercializing the improved energy-saving lamps indicates that they have potential for flowering induction in other crops and plants, as well as for alleviating the power burden in pitaya growing regions.

## Nighttime supplemental light duration and timing

In addition to light intensity, light quality, and photoperiod, the timing and duration of nighttime supplemental light are also critical technical parameters that have a profound impact on the growth of pitaya fruit. Nighttime supplemental light duration and timing can greatly influence pitaya's growth and flowering. Different lighting timings and

durations have diverse impacts. According to the articles, midnight lighting (22:30–2:30) is optimal for flower bud differentiation and yield. Generally, 4–6 h of dark-period interruption can induce flowering, but over-flowering may occur. In late autumn, a 1-h dark-period interruption is recommended for the shortest time to 50% flowering, while in early spring, 4 h is better. When it comes to nighttime supplemental light duration and timing. The control strategy includes two photoperiod control goals: flowering control and photosynthesis, while determining the timing of LED supplemental light (daytime or nighttime lighting). The nighttime lighting period can be divided into three options: the first half of the night, early morning, and late night. In Xishuangbanna, the supplemental light time for pitaya fruit is from 19:00 to 22:00, and this lighting technique not only achieves the regulation of the fruiting period according to market demand to ensure the harvest time but also meets the economic goals of high yield and high value (*Liu et al., 2019*). *Tran, Yen & Chen (2015)* found that under supplemental light, all 23 genotypes of red-pink flesh pitaya showed positive flowering responses. The duration of supplemental light required to successfully induce flowering ranged from 33 to 48 days, and the response to off-season artificial lighting induction varied by pitaya genotype in southern Taiwan, influenced by temperature. Red pitaya was more easily induced to flower by nighttime lighting than white pitaya, with off-season fruits being larger and having higher total soluble solids content (*Tran, Yen & Chen, 2015*). *Meng & Kramer (2024)* suggested that nighttime low-intensity (2 μmol·m$^{-2}$·s$^{-1}$) photoperiod lighting typically promotes flowering in long-day greenhouse ornamental plants under conditions of short natural daylengths (*Meng & Kramer, 2024*). *Gan (2022)* studied the effects of LED nighttime supplemental light on pitaya, with spring lighting from 18:00–22:00 and autumn lighting from 19:00–23:00. The results showed that LED supplemental light significantly promoted flower bud differentiation, increased the number of flower buds, and improved yield (*Gan, 2022*). *Xu et al. (2023)* used pitaya as the test material and applied red, blue, and green light quality (Red:Green:Blue=23:75:2) for supplemental light treatments, studying the effects of different lighting periods on flower bud differentiation, flowering branch rate, fruit growth, photodevelopment, and yield. The results showed that the best flower bud differentiation in pitaya occurred with the midnight lighting treatment (22:30–2:30), followed by evening lighting (18:30–22:30), with the worst results from early morning lighting (2:30–6:30). Flowering branch rate and single batch yield were also highest with midnight lighting, followed by evening lighting, with early morning lighting being the least effective. The supplemental light treatment promoted earlier flowering, with one batch of flowers blooming 15 days earlier than the control group (CK) that did not receive supplemental light. The midnight lighting treatment had the most significant effects on flower bud differentiation, flowering branch rate, and yield. *Rashid, Azam & Chowdhury (2021)* suggested that 100 W IB supplemental light for 6 h (18:00–24:00) is the optimal supplemental light method for off-season production of pitaya in Bangladesh.

In the plant life cycle, flowering marks the transition from vegetative growth to reproductive growth. Most pitaya flowers in China bloom between May and October. However, starting in November, due to insufficient sunlight, pitaya in Hainan Island stops flowering in winter, severely affecting the winter yield. Currently, although supplemental

light measures are used to promote flowering, they lack theoretical support, and their application is somewhat blind, leading to wasted electrical resources. *Xiong (2019)* used pitaya as the experimental material, statistically analyzing flowering numbers at three different time points to explore the effects of different spectra and lighting durations on pitaya flowering, and derived the optimal supplemental light measures for pitaya in winter in Hainan. The results indicated that both the timing and spectrum of supplemental light have significant main effects on inducing flowering in pitaya, with a notable interaction between the two. From early February to late February, it is recommended to use a 15 W combined light with a lighting duration of at least 5 h. From late February to early March, it is recommended to use a 15 W combined light with 4 h of lighting. From early March to the end of March, it is suggested to use a 15 W combined light with 3 h of lighting.

Nighttime supplemental light, end-of-day (EOD) lighting, and continuous day-night supplemental light are suitable for sunny and cloudy conditions, respectively. The primary difference among these three lighting methods lies in the duration of supplemental light. Typically, continuous day-night supplemental light requires a longer duration than nighttime lighting and EOD lighting. Flowering in pitaya can be induced by artificial lighting through the use of nighttime dark-period interruption (night-breaking, NB). To ensure abundant flowering, a 4–6 h dark-period interruption is often applied, but this can lead to over-flowering. *Jiang (2020)* compared four different supplemental light durations: 4 h (22:00–2:00), 2 h (23:00–1:00), 1 h (23:30–0:30), and 0.5 h (23:45–00:15) to study their effects on flower bud formation during late autumn and early spring. In late autumn, as the NB duration increased, the cumulative number of flower buds also increased, while a 0.5-h treatment was insufficient to induce 50% of buds to flower. With 1 to 4 h of NB treatment, the time to 50% flowering was 35 days. In early spring, the response of buds to NB treatment was stronger, and the duration had less impact on the time to 50% flowering. However, as the NB duration increased, the time to 50% flowering decreased, with the shortest time being 28 days in the 4-h treatment. Therefore, to achieve the shortest time to 50% flowering, it is recommended that the NB duration be 1 h in late autumn and 4 h in early spring.

## LED three-dimensional supplemental light technology for pitaya

The LED three-dimensional supplemental light for pitaya fruit is an important technological strategy and innovation that utilizes limited luminous flux to increase the light exposure and orientation within the pitaya canopy. Traditional top-down unidirectional lighting often results in uneven illumination, shading, and low light utilization, leading to faster senescence of leaves at the bottom of the canopy and reduced fruit production inside the canopy. To address these issues, we propose LED three-dimensional supplemental light, a multi-directional combination lighting method. Based on top canopy LED supplemental light, multiple supplemental light methods are employed, including between the canopy layers, below the canopy, and on the sides of the canopy. LED three-dimensional supplemental light offers significant advantages, such as a uniform distribution of light throughout the canopy, all-round plant growth, and high light utilization efficiency. Pitaya's fleshy branches bend downward, and its

three-dimensional growth is evident, so traditional top LED lighting cannot penetrate the canopy, leading to significant shading and uneven light intensity across the canopy. The light intensity at the top of the canopy is much higher than at the middle and lower sections, and light inside the canopy is very weak, negatively affecting the number of flowers and fruit set in the middle and lower parts of the canopy. Therefore, it is essential to establish a three-dimensional LED lighting model, comprising Mode 1 (top lighting + reflective film (*Zhang et al., 2019*) on the ground bed), Mode 2 (top lighting + bottom upward supplemental light on the ground bed), and a combination of side lighting + reflective film on the ground bed. Considering the cost of installation and equipment procurement, Mode 1 is more economical. The specific technical parameters are as follows: LED fixtures are suspended 50 cm above the pitaya plants, with a fixture spacing of 150 cm, a luminous angle of 120°, and a cut-off lens for light distribution. The lighting covers the top of the canopy and the sides of the canopy on adjacent beds. The cultivation bed height is <50 cm, with a width of 1 meter, covered with reflective film, while the aisles remain bare. Adequate lighting conditions for early flowering, fruit setting, high-quality pitaya fruit production, early color change, and early harvest are critical for achieving high yield and quality. However, in winter and spring, pitaya cultivation faces challenges such as weak light intensity, short daylight duration, uneven light distribution, and lack of light quality regulation, which need to be addressed. It is recommended that LED artificial supplemental light, combined with reflective film on the ground, form the primary and passive regulation methods to establish a three-dimensional light environment control system. Further research on the synergistic effects of LED artificial lighting and reflective film on the ground is needed.

## CONCLUSION AND PROSPECTS

Against the backdrop of increasing demand for healthy food, pitaya has become a highly favored green health food due to its unique nutritional value and excellent quality. pitaya, as a plant that prefers sunlight for long periods of time, theoretically has the potential for annual production. However, in the actual off-season cultivation process of winter and spring, pitaya faces problems such as low flowering and fruit setting rates, and difficulty in achieving ideal levels of fruit yield and quality. After research, it has been found that weak light environment and insufficient light time have become key technical bottlenecks restricting the off-season production of pitaya industry. In order to achieve multiple crop yields and high quality of pitaya, implementing LED artificial light supplementation in winter and spring seasons to induce a cascade of high-quality flowering is an essential engineering and technical measure to ensure annual efficient production. Among various supplemental light techniques, interrupting LED supplemental light during the dark period at night has become the main way to induce flowering. Based on this, further exploration can be conducted to establish a three-dimensional precise supplemental light technology. Currently, there is relatively little research on the optimal light intensity, light quality, light cycle, light distribution, and supplemental light technology parameters for LED supplemental light. LED supplemental light does promote the pitaya flowering in number and time, fruit yield and quality together. Red, blue, and far-red light, also

photoperiod of 4–6 h are recommended for application in supplemental light. The depth and breadth of research in this field need to be further expanded. The future research and development direction should focus on establishing a pitaya LED supplemental light technology with dark period interruption as the core, based on in-depth understanding of the physiological regulation mechanisms of light intensity, light quality, and light cycle. A matching three-dimensional precise supplemental light strategy should be constructed, and specialized spectral and optical supplemental light fixtures should be customized and developed. In summary, in-depth research on the effect and physiological mechanism of LED supplemental light induced flowering, the development of LED supplemental light with precise spectrum and light distribution, and the establishment of precise three-dimensional supplemental light technology based on this will be the core direction of future pitaya supplemental light research. This will not only help break through the current technological bottleneck in the development of the pitaya industry and achieve efficient production every year, but also inject new vitality into the sustainable development of the pitaya industry.

### Funding
This work was supported by the Research Fund of Guangdong-Hong Kong-Macao Joint Laboratory for Intelligent Micro-Nano Optoelectronic Technology (No. 2020B1212030010). The funders had no role in study design, data collection and analysis, decision to publish, or preparation of the manuscript.

### Grant Disclosures
The following grant information was disclosed by the authors:
Guangdong-Hong Kong-Macao Joint Laboratory for Intelligent Micro-Nano Optoelectronic Technology: 2020B1212030010.

### Competing Interests
The authors declare that they have no competing interests.

### Author Contributions
- Ren Chen conceived and designed the experiments, performed the experiments, analyzed the data, prepared figures and/or tables, authored or reviewed drafts of the article, and approved the final draft.
- Yiming Ding performed the experiments, prepared figures and/or tables, authored or reviewed drafts of the article, and approved the final draft.
- Wenke Liu conceived and designed the experiments, analyzed the data, prepared figures and/or tables, authored or reviewed drafts of the article, and approved the final draft.
- Xianwei Zhan analyzed the data, authored or reviewed drafts of the article, and approved the final draft.
- Kexin Lin analyzed the data, prepared figures and/or tables, authored or reviewed drafts of the article, and approved the final draft.
- Kaifeng Lian performed the experiments, prepared figures and/or tables, authored or reviewed drafts of the article, and approved the final draft.
- Weilong Chen analyzed the data, prepared figures and/or tables, authored or reviewed drafts of the article, and approved the final draft.
- Keyu Wang performed the experiments, authored or reviewed drafts of the article, and approved the final draft.
- Shangfei Lin conceived and designed the experiments, authored or reviewed drafts of the article, and approved the final draft.

## Data Availability

This is a literature review.

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
