# Peer review of "Physiological effects and technical strategies of LED supplemental lighting for pitaya cultivation: a review"

_PeerJ, doi:10.7717/peerj.19720_

## Round 0.1 · original submission · Major Revisions

The reviewers have made some suggestions that will add value to your review.

·

Basic reporting

This article presents a review with the objective of analyzing the existing research results, thus clarifying the current situation of LED supplementary light's physiological effects and technology strategy in pitaya cultivation. Literature references are appropriate. There are no tables or figures in the article. For a review article, it would be interesting to have tables to summarize the results and figures to illustrate them. The field of this review is interesting.

Experimental design

The primary retrieval tools for this research were Google Scholar and the China National Knowledge Infrastructure (CNKI) database. Consulting the Web of Science databases could improve the scope of the review. Information about intra- and interspecific genetic variability could be added to the review article. Do all commercial pitaya species and varieties have the same physiological behavior?

Validity of the findings

The review on the subject is interesting, considering that pitaya cultivation is growing in several countries and that light supplementation is a technology that has an impact on the production sector. More specific and objective recommendations on light intensity, light quality, light photoperiod, and technical strategies for inducing flowering could be included in the conclusions.

Additional comments

Considering the suggestions in the above items, the article can be accepted for publication.

Reviewer 2 ·

Basic reporting

The manuscript should ensure clarity and cohesiveness in its structure. While the sections are outlined, some areas need clearer transitions between subsections to enhance readability. A more defined introduction that sets a clear research question would strengthen the paper. The review could benefit from a more comprehensive synthesis of recent studies in the field. Although some references are provided, the integration of more diverse viewpoints and studies could address potential biases and gaps in the existing literature.

Experimental design

The paper demonstrates potential and has significant relevance to the field of agricultural science, specifically regarding LED lighting in horticulture. However, to be deemed acceptable, the authors should address the comments above to enhance clarity, depth, and coherence. If these suggestions are implemented, it would likely improve the manuscript's strength and its potential for acceptance. In summary, while the foundational aspects of the review are strong, there is a clear need for revisions, particularly in the areas of depth of discussion, clarity, and the presentation of the findings. If the authors can effectively address these points, the paper could be considered for publication.

Validity of the findings

The paper identifies certain gaps but does not fully explore potential methodologies for future research, such as the use of varied environmental conditions or different cultivars in their studies. An expanded discussion on innovative research methodologies could strengthen the manuscript. The readers may require additional technical details regarding how specific LED light conditions impact physiological responses at the biochemical level. Providing more empirical data, where possible, can help substantiate claims made about the physiological effects of light conditions on plant growth. The suggestion for cross-disciplinary collaboration is a strong point. However, it could be more explicitly detailed how these collaborations could economically and practically affect the industry's sustainability and efficiency.

Additional comments

1. Lines 1-2 (Abstract):

- The abstract lacks a concise summary of key findings or conclusions. It should include specific results or implications of LED lighting on Pitaya cultivation.

- Suggestion: Revise the abstract to summarize key findings succinctly, including physiological effects and practical applications.

2. Lines 118-120:

- The statement about international research progress is quite broad. It lacks specificity regarding the contributions of the cited works to the topic of LED lighting in Pitaya.

- Suggestion: Include more detail about how Shah et al. (2023) and Oliveira and Tel Zur (2022) specifically relate to LED lighting in Pitaya cultivation.

3. Lines 135-136:

- The audience identification is solid, but this section could benefit from integrating how the findings can directly influence each identified group.

- Suggestion: Elaborate on the potential applications for each audience group to enhance relevance.

4. Lines 149-151:

- The inclusion criteria are stated well, but it would be helpful to clarify why the exclusion of non-Pitaya studies is justified.

- Suggestion: Add a brief rationale for focusing specifically on Pitaya to support the relevance of your review.

5. Lines 170-179:

- The identification of gaps is insightful; however, the need for further research is mentioned a bit superficially without concrete examples.

- Suggestion: Provide examples of potential experiments or studies that could fill these gaps, such as specific LED settings for different growth stages.

6. Lines 185-191:

- The description of light intensity categories is informative, but could be better integrated with how these categories directly affect Pitaya specifically.

- Suggestion: Specify the correlation between these categories and Pitaya's growth parameters or yield metrics.

7. Lines 562-564:

- The recommendation for reflective film usage alongside LED lighting is an interesting proposition, but lacks evidence or previous studies to support this combination.

- Suggestion: Cite any relevant studies or data that support this claim to lend credibility to this suggestion.

8. Lines 574-577:

- The conclusion reiterates the issues faced in the industry well, but it would be beneficial if it shared more definitive next steps or a call-to-action for researchers.

- Suggestion: Clearly outline practical applications or recommendations based on your findings that could be useful for industry stakeholders.

---

## Round 0.2 · accepted · Accept

Dear Authors,
The Reviewer has made the review of your manuscript titled "Physiological effects and technology strategy of LED supplemental light for Pitaya cultivation: a review." Based on that opinion, I am pleased to inform you that you have performed substantial and meaningful revisions in response to the initial feedback. The article has been significantly improved in terms of clarity, structure, and scientific rigor.
Your article can be published in its present form, my congratulations!

Reviewer 2 ·

Basic reporting

No comment

Experimental design

No comment

Validity of the findings

No comment

Additional comments

No comment